# Urban Environmental Quality Assessment by Spectral Characteristics of Mulberry (*Morus* L.) Leaves

**Snejana Dineva** [1,2]**, Petya Veleva-Doneva** [1,2] **and Zlatin Zlatev** [1,2,*]

[1]   Faculty of Technics and Technologies, Trakia University, 8602 Yambol, Bulgaria;
      snezhana.dineva@trakia-uni.bg (S.D.); petya.veleva@trakia-uni.bg (P.V.-D.)
[2]    Department of Agricultural Engineering, Faculty of Agriculture, Trakia University, 6000 Stara Zagora, Bulgaria
[*]   Correspondence: zlatin.zlatev@trakia-uni.bg

**Abstract:** In this paper, an analysis of the possibility of passive determination of the degree of environmental pollution based on data from the leaf blade of mulberry is made. With existing solutions in this area, the mulberry has been found to be under-researched. A disadvantage of the available solutions is that spectral indices are used, which is not a sufficient criterion for passively determining the degree of air pollution based on the surface characteristics of the mulberry leaves. Methods have been used to reduce the amount of data by latent variables and principal components. It has been found that a kernel variant of the principal components, combined with linear discriminant analysis, is an appropriate method for distinguishing the degree of air pollution from mulberry leaf data. The results obtained can be used to refine the approaches used to passively determine the degree of air pollution in the habitat area of the plant. Methods and software tools could be used to develop mobile applications and new approaches to remote sensing, in express determination of the degree of environmental pollution, according to data from the mulberry leaves.

**Keywords:** passive biomonitoring; mulberry leaves; PCA; discriminant analysis; spectral indices

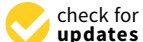



## 1. Introduction

Environmental quality monitoring in urban areas is a method that offers the opportunity to avoid adverse effects on human health. Passive and active biomonitoring of air quality has both advantages and disadvantages. Passive biomonitoring has the advantage of using tree species already present in the ecosystem, making this approach affordable and effective over time [1].

Detailed studies have been conducted on passive bio-monitoring of air quality based on leaf data from Tilia (*Tilia* sp.) [2], hornbeam (*Carpinus betulus*) [3], and white willow (*Salix mucronata*) [4]. In the studies of these plants, both classical laboratory methods and contactless measurement techniques were used, such as spectral characteristics in the visible and near infrared, as well as hyperspectral analysis. The requirements for the application of different types of plants to passively determine the quality of the air in cities include identifying features of pollution that alter plant characteristics, needing to develop new methods, or refining existing ones [5–9]. The aim of this study is to address precisely these requirements.

A plant that has a proven content of bioactive substances [10], is resistant to contaminated soils, and allows pesticide treatment [11] is the mulberry. Additionally, the plant is sensitive to changes in the environment [12]. The economic importance of mulberries is to feed silkworms, cattle, and goats. In some cases, it is used as a park tree. In Bulgaria it is mainly found near major roads in the cities. The wide use of mulberry for creating plantations for different purposes (leaf mass, fruit, wood, landscaping), and the versatile application that the individual parts of the tree have, determine its great economic importance.

Many of the published results related to mulberry analysis and evaluation, include spectral and hyperspectral analysis methods, to define different qualitative indicators such as proteins, vegetable oils, starch, dry matter, moisture, acidity, various mycotoxins, and infections affecting indicators, relating to the application of the plant for medical and business purposes. For the most part, the laboratory methods used for analyzing mulberries are subjective and require considerable time to process plant samples. The accuracy of diagnosis is not high and depends on the expert's qualifications. That is why the creation of highly efficient automated technologies for the evaluation of mulberry indicators is a priority objective of current research in this field. The purpose of spectral methods is to speed up the process of determining the state of the environment, not to replace classical laboratory methods.

Directive 2008/105/EC determines the good chemical status to be achieved by all Member States of the European Union (including Bulgaria), the legal basis for monitoring priority substances in sediments and flora and fauna. The laboratory and field methods described in this normative document include determination of $O_2$, $CO_2$, CO, SOx, NOx, $Cl_2$, $H_2S$, HCl, VOC, and PM in the air. Additionally, it describes determining the content of heavy metals, anions, and pesticides.

There are few studies on the qualitative indicators related to the surface texture of mulberry leaves, as well as the influence of the polluted environment in the habitat area of the plant. The question of whether the mulberry is suitable for passive biomonitoring for air quality remains unclear.

The purpose of this study is to analyze changes in the spectral reflectance characteristics of mulberry leaves depending on the level of air pollution in the habitat area of the plant.

## 2. Material and Methods

The mulberry leaf samples were taken from six areas with high and low car traffic. 25 leaves were used from each area taken from the sun-exposed side of the trees. The leaves were transported in a cooler bag. The measurements were made as soon as they were delivered to the laboratory.

To determine the environmental parameters in the analyzed areas, an experimental set-up was used, developed at the Faculty of Technics and Technologies, Yambol, Bulgaria [13]. The measuring device consists of a sensor module and a microprocessor control system offering wireless communication.

By the system were measured: smoke gasses, ppm; particle matter PM > 0.5 $\mu g/m^3$; equivalent $CO_2$, ($eCO_2$), ppm; and total volatile organic compounds, TVOC, ppb.

The measurements were made at a temperature of $22 \pm 3\ °C$ and a relative humidity of $39 \pm 5\%$ RH.

Table 1 shows the data on the areas in which the leaves were taken. Pollution rates and geographical coordinates are indicated. The area is located in the southeastern part of Bulgaria. Mulberries are more common urban plants in the studied geographical region.

**Table 1.** Areas for collecting mulberry leaves.

| Zone | Degree of Air Pollution | Geographic Coordinates (WGS 84) |
|------|-------------------------|--------------------------------|
| Z1 | P | 42°28′44.20″ N; 26°31′24.19″ E |
| Z2 | P | 42°28′48.29″ N; 26°30′29.18″ E |
| Z3 | P | 42°28′55.27″ N; 26°30′15.23″ E |
| Z4 | LP | 42°15′25.34″ N; 26°37′35.29″ E |
| Z5 | LP | 42°15′26.24″ N; 26°37′35.71″ E |
| Z6 | LP | 42°28′40.75″ N; 26°30′25.27″ E |

LP—less polluted; P—polluted.

The leaves are grouped into 2 groups—derived from less polluted zones (LP) and polluted zones (P). Passive determination is equal to passive biomonitoring in this case.

### 2.1. Measurement of Air Environmental Parameters

To determine the environmental parameters in the analyzed zones, an experimental set-up was used, developed at the Faculty of Technics and Technologies, Yambol, Bulgaria [13]. The measuring device consists of a sensor module and a microprocessor control system offering wireless communication. The system measured smoke gasses, ppm; particle matter $PM > 0.5 \ \mu g/m^3$; equivalent $CO_2$, $eCO_2$, ppm; and total volatile organic compounds, TVOC, ppb. The measurements were made at a temperature of $22 \pm 3 \ ^\circ C$ and a relative humidity of $39 \pm 5\%$ RH.

### 2.2. Planar Chromatography

The method used was that presented in Priyadarshini et al. [14], with some modifications. The mulberry leaves from the polluted and less polluted areas were cut into $10 \times 10$ mm pieces. Their handles were removed. They were soaked for 4 h in acetone. On a white paper with a density of $80 \ g/m^2$ and dimensions $105 \times 19$ mm, a drop of extract was applied at a distance of 15 mm from the end of the paper. The sample was dried for 15 min. Four mm from the end of the paper was immersed in acetone. After 15 min, the samples were removed from acetone and dried for 1 h. The values of the 5 separated fractions were then reported. Three replicates were made and the mean and standard deviation of Rf were reported. Rf = A/B, where A is the distance recorded by the solvent, and B is the distance reached by the corresponding fraction. Carotene, xantophyll, chlorophyll a, chlorophyll b, and anthocyanin fractions are reported.

### 2.3. Determination of Physicochemical Parameters

The preparation of the measurement samples was carried out according to the procedure presented in AACC 02-52.01 [15], with some modifications suitable for the electrometric measurement of leaf parameters, in the following order: distilled water was heated to $70 \ ^\circ C$; the leaf mass was crushed and placed in distilled water at a ratio of 1/10 (5 g of raw material in 50 mL of distilled water); stirring; and after cooling to ambient temperature, 3 consecutive measurements of each indicator were made and their average value and standard deviation were determined.

Measuring instruments used: Technical balance MH-200 (ZheZhong Weighing Apparatus Factory, Yongkang, China), maximum defined mass 200 g, with a resolution of 0.02 g; active acidity pH, pH meter PH-108 (Hangzhou Lohand Biological Co., Ltd, Hangzhou, China); EC conductivity, $\mu S/cm$, Conductivity Meter AP-2 (HM Digital, Inc., Redondo Beach, CA, USA); total dissolved solids, ppm, TDS-3 measuring instrument (HM Digital, Inc., Redondo Beach, CA, USA); and redox potential ORP, mV, Measuring Instrument Model ORP-2069 (Shanghai Longway Optical Instruments Co., Ltd, Shanghai, China).

### 2.4. Experimental Set-Up for Obtaining Spectral Characteristics

The experimental set-up used consists of a personal computer with software for receiving and processing images and spectral characteristics in the visible and near-infrared areas. The spectral characteristics of leaves were captured with a spectrophotometric sensor TCS230 (TAOS Inc., Premstaetten, Austria). The sensor is controlled by single-board microcomputer Arduino Nano compatible (Kuongshun Electronic Ltd., Shenzhen, China). The measuring distance was 0.5 cm from the leaf to the sensor. White LEDs with a maximum light intensity of 450 nm were used. The measurements are for 5 points of the adaxial and also 5 points on the abaxial part of the leaves.

### 2.5. Obtaining Spectral Characteristics

The transformation of values from *XYZ* and *LMS* models into reflection spectra in the VIS and NIR, in the 390–730 nm and 800–1000 nm ranges, was performed mathematically

and the transformation was possible in both directions of equality [16]. Mathematical dependencies, with the possibility of converting in both directions of equality:

$$X = \int_{\lambda_1}^{\lambda_2} A(\lambda)\overline{X}(\lambda)d\lambda; \; Y = \int_{\lambda_1}^{\lambda_2} A(\lambda)\overline{Y}(\lambda)d\lambda; \; Z = \int_{\lambda_1}^{\lambda_2} A(\lambda)\overline{Z}(\lambda)d\lambda \tag{1}$$

where $A(\lambda)$ is a matrix for converting color to reflection spectra in the VIS range, for accepted observer and illumination.

The used matrices for converting (matching functions) color components to spectrum are available in [17] for the VIS region. Conversion functions for observer 2° are applied (LMS 2°, CIE 2006).

The conversion to NIR was performed using the compliance functions presented in [18]. The illumination data used to convert the VIS and NIR characteristics were in accordance with D65 (average daylight with UV component (6500 K)) illumination. The conversion function between the *RGB* and *XYZ* models, in the range $\lambda_1$–$\lambda_2$ (380–780 nm), can be represented as:

$$XYZ = RGB \cdot M$$
$$M = \begin{bmatrix} 0.5767 & 0.2974 & 0.0270 \\ 0.1855 & 0.6273 & 0.0707 \\ 0.1882 & 0.0753 & 0.9911 \end{bmatrix} \tag{2}$$

where $M$ is the transformation matrix under the specified conditions for observer 2° and illumination D65. From here, the spectral characteristic is of the form:

$$S_{\text{VIS}} = \sqrt{\Delta X^2 + \Delta Y^2 + \Delta Z^2} \tag{3}$$

Conversion functions change the way spectral data is stored or the way that it is represented. The conversion function in the range $\lambda_1$–$\lambda_2$ (800–1000 nm) between the *XYZ* and the *LMS* model can be represented as:

$$LMS = XYZ \cdot T$$
$$T = \begin{bmatrix} 0.7328 & 0.4296 & -0.1624 \\ -0.7036 & 1.6975 & 0.0061 \\ 0.0030 & 0.0136 & 0.9834 \end{bmatrix} \tag{4}$$

where $T$ is the transformation matrix under the specified conditions for observer 2° and illumination D65. From here the spectral characteristic is of the form:

$$S_{\text{NIR}} = \sqrt{\Delta L^2 + \Delta M^2 + \Delta S^2} \tag{5}$$

### 2.6. Determination of Information Indices by Spectral Characteristics

The *NDAI* (Normalized Dorsiventral Asymmetry Index) is defined as a linear combination of the reflections of the adaxial and abaxial parts of the leaves.

$$NDAI = \frac{\rho_{I,ab} - \rho_{I,ad}}{\rho_{I,ab} + \rho_{I,ad}} \tag{6}$$

where $\rho_{I,ab}$ is the reflection of the abaxial part of the leaf; $\rho_{I,ad}$—reflection of the adaxial part of the leaf.

The two types of reflection of the leaves are determined in the same wavelength. The blue part of the visible spectrum is at 420 nm, the green at 520 nm, and the red at 620 nm.

### 2.7. Reducing the Amount of Spectral Characteristics Data

Latent variables (LV), principal components (PC), and kernel variant of principal components (kPC) were used to reduce the amount of spectral characteristics data [19]. The kernel version uses three kernel functions: Simple; Polynomial; and Gaussian. Software tools described by Wang [20] were used to obtain the kernel principal components.

The PCA kernel method can be summarized in the following steps:

✓   Creating a *K* kernel matrix from the training sample {$x_i$} by:

$$K_{i,j} = k(x_i, x_j) \tag{7}$$

✓   Gram $K'$ matrix calculation:

$$K' = K - 1_N K - K 1_N + 1_N K 1_N \tag{8}$$

✓   Calculating vectors a$_i$ by dividing *K* by $K'$:

$$K a_k = \lambda_k N a_k \tag{9}$$

✓   Calculation of kernel principal components $y_k(x)$:

$$y_k(x) = \Phi(x)^T v_k = \sum_{i=1}^{N} a_{ki} k(x, x_i) \tag{10}$$

### 2.8. A Correlation Method Was Used

This method determined the strength of the relationship between the *NDAI* spectral index and the physicochemical characteristics of mulberry leaves. The distribution of the data was checked by the methods: Shapiro–Wilks test; Kolmogorov–Smirnov test; and Lilliefors test.

As a criterion for evaluation, a correlation coefficient R was used. At R < 0.3, there is no or a very weak relationship between the data; at 0.3 < R < 0.5, the relationship is weak; at 0.5 < R < 0.7 the relationship is moderate; and for R > 0.7 the relationship is strong.

### 2.9. Classification Methods Used

The Naïve Bayes classifier was used as a reference [21,22]. One of the classic algorithms in machine learning is the Naïve Bayes Classifier, which is based on the Bayes theorem for determining the posterior probability of an event occurring. Accepting the "naïve" assumption of conditional independence between each pair of attributes, the Naïve Bayes classifier effectively handles too many attributes to describe an example, i.e., with the so-called "The curse of dimension". Bayes's theorem:

$$P(y = c|x) = \frac{P(x|y = c)P(y = c)}{P(x)} \tag{11}$$

where $P(y = c \,|\, x)$ is the probability of an object belonging to a class c (posterior probability of the class); $P(x \,|\, y=c)$—the probability of the object *x* to meet in the middle of the object of class c; $P(y = c)$—unconditional probability of occurrence of object *y* in class *c* (a priori probability of class); and $P(x)$—unconditional probability of object x.

The purpose of the classification is to determine to which class the object *x* belongs. Therefore, it is necessary to find the probability class of the object *x*, i.e., it is necessary for all classes to select the one that gives the maximum probability $P(y = c \,|\, x)$.

$$c_{opt} = \underset{c \in C}{argmax} \ P(x|y = c)P(y = c) \tag{12}$$

The definition of boundary values for the separation of polluted and less polluted zones, depending on the characteristics of the mulberry leaves, was made by discriminant analysis using a linear separation function (LDA) [23]. LDA is suitable for datasets that have high clustering and low variance. In general, the linear separating function is:

$$\delta_k(x) = x^T \Sigma^{-1} \mu_k - \frac{1}{2} \mu_k^T \Sigma^{-1} \mu_k + \log(\pi_k) \tag{13}$$

where $\delta_k$ is a separating function; $\mu_k$ is the average vector; *x*—observations; and $\Sigma^{-1}$—covariance matrix. For practical purposes, it is convenient to present the separation function as:

$$\delta(v) = K + v \cdot L \tag{14}$$

where *K* is a constant; *L*—linear coefficient; and *v* = [*x;y*]—vectors (matrix) of the data *x* and *y*.

Among use of the classifiers, one of the most important parts of the work is the choice of an appropriate measure in order to properly assess the classification performance. The evaluation of the performance of the classifiers used is based on a general classification error, which is described by the formula:

$$\text{e} = \frac{\sum_{i=1}^{n}(\sum_{k=1}^{n} y_{ik} - y_{ii})}{\sum_{i=1}^{n}\sum_{k=1}^{n} y_{ik}} \cdot 100 \; in \; \% \qquad (15)$$

where $y_{ik}$ is the number of class *i* samples classified by classifier in class *k*; $y_{ii}$—number of correctly recognized samples; $k = 1, \ldots, n$—number incorrectly assigned to a class *i* relative to the total number of samples; and *n*—number of classes. All data were processed at a level of significance of $\alpha = 0.05$.

## 3. Results and Discussion

Effective application of mulberry leaves data to determine the degree of pollution of the habitat area is entirely aimed at using methods that would be sufficiently effective with respect to rapid and simple classification, and at the same time giving satisfactory accuracy according to generally accepted standards to that end.

The results presented can be summarized in three groups. In the first stage, technological measurements of the mulberry leaves were made, including chromatographic and physicochemical methods of analysis. In the second stage, data from the analysis of spectral characteristics in the visible and near infrared spectral ranges are presented, both in their direct use and by methods of the amount of data reduction and classification. Finally, a discussion is made in which the results obtained are compared with those reported by other authors.

The measured parameters of the environment in the habitats of mulberry are shown in Table 2. It was found that in the high-pollution areas, the parameter values were significantly higher than in the low-pollution zones.

**Table 2.** Mean annual pollutants concentration for polluted (P) and less polluted (LP) areas.

| Parameter / Zone | Smoke Gasses, ppm | PM > 0.5 $\mu g/m^3$ | eCO$_2$, ppm | TVOC, ppb |
|---|---|---|---|---|
| P | $0.57 \pm 0.1$ | $79.32 \pm 0.6$ | $641 \pm 37$ | $35 \pm 2$ |
| LP | $0.06 \pm 0.002$ | $12.36 \pm 1.4$ | $422 \pm 12$ | $3 \pm 0.3$ |

Table 3 presents the results of planar chromatography on mulberry leaves from polluted (P) and less polluted (LP) areas. It is seen that for the leaves of the polluted areas, the values of the individual parameters are significantly lower than those of the less polluted areas. It is also seen that the coefficient of variation (CV) is below 30% (CV = SD/mean).

**Table 3.** Results from planar chromatography.

| Component / Zone | Carotene | Xantophyll | Cholophyll a | Chlorophyll b | Anthocyanin |
|---|---|---|---|---|---|
| P | $0.67 \pm 0.11$ | $0.53 \pm 0.15$ | $0.45 \pm 0.14$ | $0.26 \pm 0.07$ | $0.09 \pm 0.02$ |
| LP | $0.75 \pm 0.08$ | $0.62 \pm 0.15$ | $0.54 \pm 0.16$ | $0.31 \pm 0.09$ | $0.14 \pm 0.04$ |

Table 4 shows the results of physicochemical parameters of mulberry leaves from polluted (P) and less polluted (LP) areas. Compared to the less polluted areas, the leaves from the polluted areas have higher values of active acidity and redox potential, lower values of electrical conductivity, and completely dissolved substances. It is also seen that the coefficient of variation (CV) is below 30% (CV = SD/mean). As in the previous cases, this indicator is higher for mulberry leaves than less polluted areas.

**Table 4.** Physicochemical characteristics of mulberry leaf.

| Characteristic / Zone | pH | EC, µS/cm | TDS, ppm | ORP, mV |
|---|---|---|---|---|
| P | $7.8 \pm 0.5$ | $450 \pm 34$ | $130 \pm 12$ | $165 \pm 12$ |
| LP | $7.5 \pm 0.9$ | $514 \pm 74$ | $139 \pm 23$ | $151 \pm 21$ |

Figure 1 shows the averaged VIS spectral characteristics for the adaxial and abaxial part of mulberry leaves. It can be seen that the adaxial part has a separation between the spectral characteristics between the leaves in the polluted and less polluted areas. Only overlap 490–510 nm is observed. There is a strong overlap in spectral characteristics at the abaxial part of the leaves. Only in the 380–500 nm range is there a visible resolution between these characteristics.

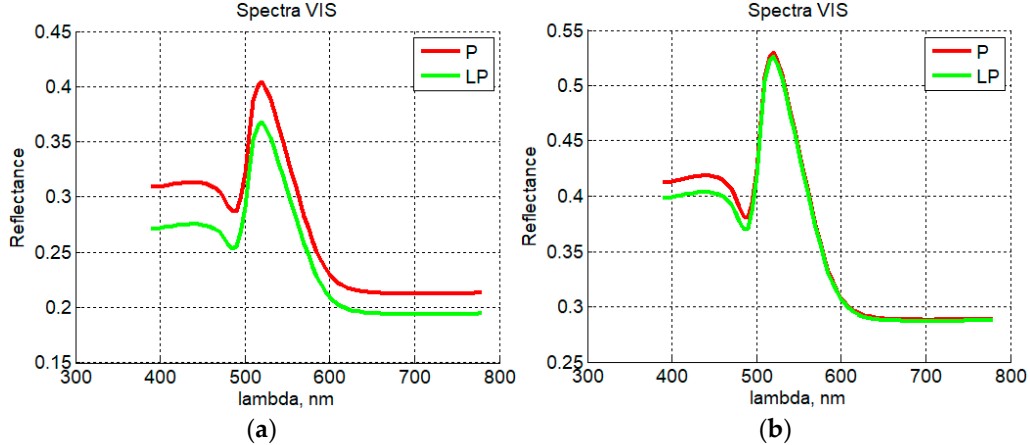

**Figure 1.** VIS spectral characteristics of mulberry leaves (**a**) adaxial part and (**b**) abaxial part.

Figure 2 shows the averaged NIR spectral characteristics for the adaxial and abaxial part of a mulberry leaf. Both parts have strong overlapping spectral characteristics. The separation is observed in the 820–860 nm ranges as well as at 880–950 nm. In the second spectral range, the separation of the characteristics is more pronounced at the abaxial part of the leaves.

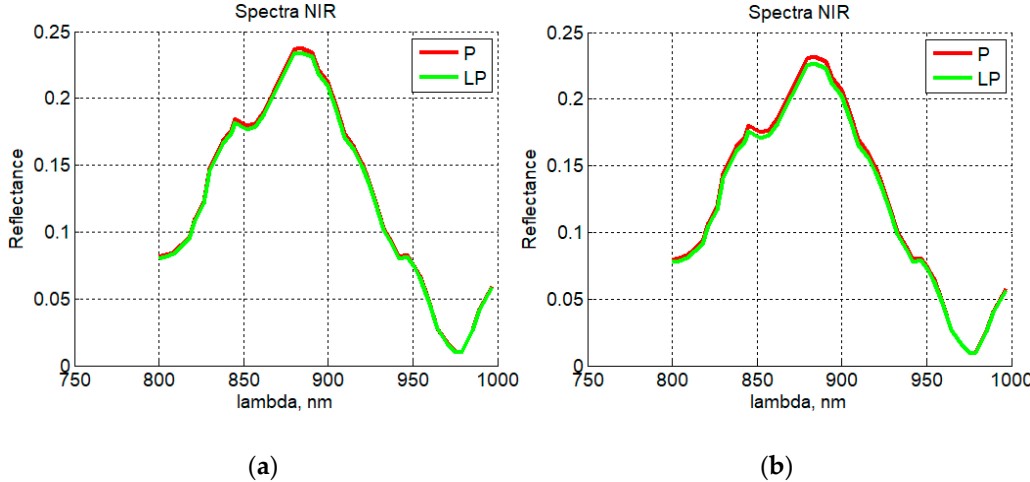

**Figure 2.** NIR spectral characteristics of mulberry leaves (**a**) adaxial part and (**b**) abaxial part.

Figure 3 presents the results for *NDAI* indices obtained from the correlation between the spectral characteristics measured from the adaxial and abaxial parts of the leaf petal. As can be seen from the figure in the mean values, there is a difference in the spectral

indices for mulberry leaves from the polluted and less polluted area. Their standard deviations overlap, which indicates that a breakdown of these indices cannot be made for all measurement cases. These results indicate that the direct use of spectral characteristics data is not appropriate in distinguishing between mulberry leaves from polluted and less polluted areas.

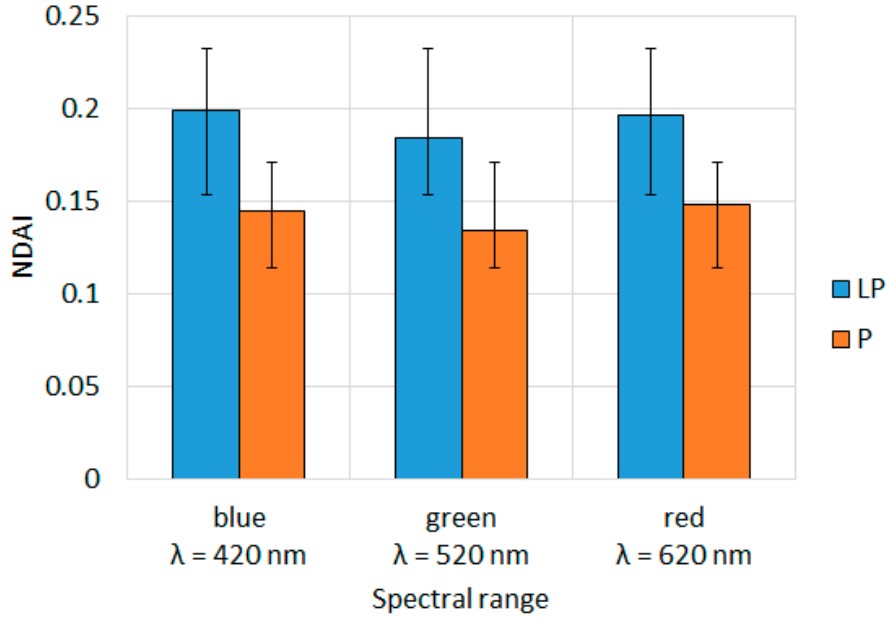

**Figure 3.** NDAI VIS spectral indices.

The correlation between the *NDAI* spectral index and the physicochemical characteristics of mulberry leaves was evaluated. From the analysis of the distribution, it was found that $p = 0.07–0.09$. At $df = 18–74$, it can be assumed that the data have a distribution close to normal.

Figure 4 shows the correlation between the analyzed values. At $\lambda = 420$ nm, corresponding to the blue color of the spectrum, a strong correlation ($R > 0.7$) of the *NDAI* index was observed with anthocyanin, pH, EC, and TDS. At $\lambda = 520$ nm, corresponding to the green color of the spectrum, a strong correlation ($R > 0.7$) of the *NDAI* index was observed with carotene, xanthophyll, chlorophyll a, and pH. At $\lambda = 620$ nm, corresponding to the red color of the spectrum, a strong correlation ($R > 0.7$) of the *NDAI* index was observed with carotene, xanthophyll, chlorophyll a, and pH.

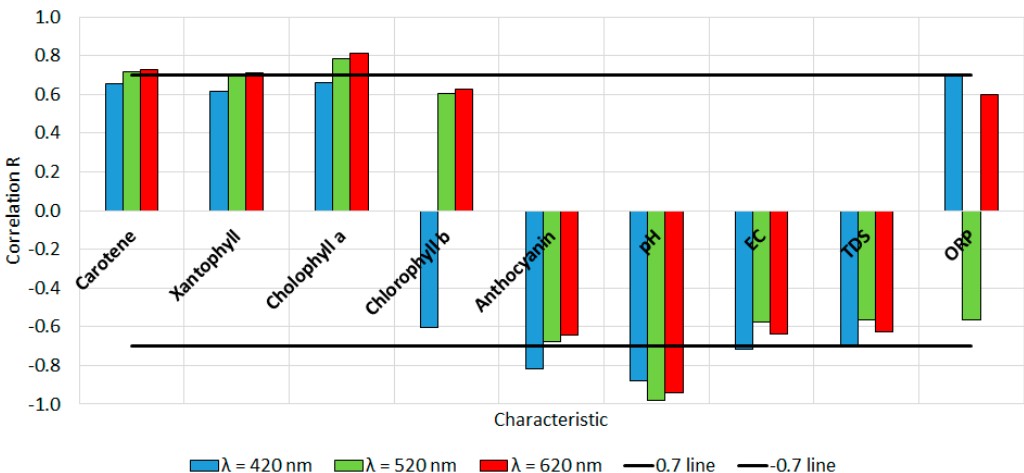

**Figure 4.** Correlation between *NDAI* spectral index and physicochemical characteristics of mulberry leaves.

The strong relationship between the *NDAI* spectral index and chlorophyll is due to the fact that they absorb light most strongly in the blue part of the spectrum, as well as in the red part. Conversely, they are a poor absorber of green and almost green parts of the spectrum, which it reflects, producing a green color to tissues containing chlorophyll [24].

The possibility of distinguishing mulberry leaves from polluted and less polluted areas was examined by using methods to reduce the amount of data of the spectral characteristics of mulberry leaves in the visible and near infrared ranges of the spectra.

Figure 5 shows an example of the work of a Naïve Bayes classifier. The results shown are using reduced spectral characteristics data from the adaxial part of the bilberry leaf, reduced by the kernel principal components method.

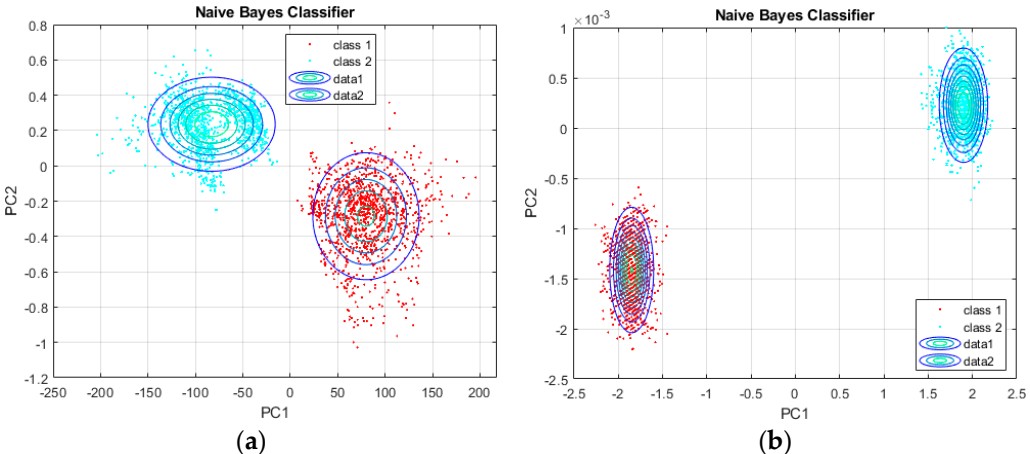

**Figure 5.** An example of a Naïve Bayes classifier work: (**a**) VIS adaxial, kPCA simple and (**b**) NIR adaxial, kPCA simple.

Table 5 shows the results of a common classification error using the accepted Naïve Bayes classifier. It can be seen that, using latent variables and the linear variant of the principal components, too-large values of the total classification error of over 45% are obtained. This is an expected result, since these two methods produce reduced values that are close in nature to the spectral characteristics.

**Table 5.** Common classification error (e, %) of the Naïve Bayes classifier.

| Method / Leaf Part Spectra | LV | PC | kPC Simple | kPC Polynomial | kPC Gaussian |
|---|---|---|---|---|---|
| VIS adaxial | 49% | 45% | 0% | 1% | 41% |
| VIS abaxial | 41% | 45% | 0% | 0% | 12% |
| NIR adaxial | 44% | 48% | 0% | 0% | 0% |
| NIR abaxial | 45% | 48% | 0% | 0% | 0% |

Significantly lower error values were obtained using the kernel variant of the principal components. Only with the use of the Gaussian kernel, for spectral characteristics in the visible spectral range, high values of the common classification error of more than 40% are obtained.

For the next stages of work, a kernel variant of the principal components using the simple and polynomial kernel functions is selected.

Figure 6 shows in general the results of using a linear discriminant classifier. It can be seen that, when using a polynomial kernel, it produces worse results than using a simple kernel. The data overlap, which is a prerequisite for increasing the classification error and hence reducing the accuracy in distinguishing polluted and less polluted areas according to mulberry leaf data.

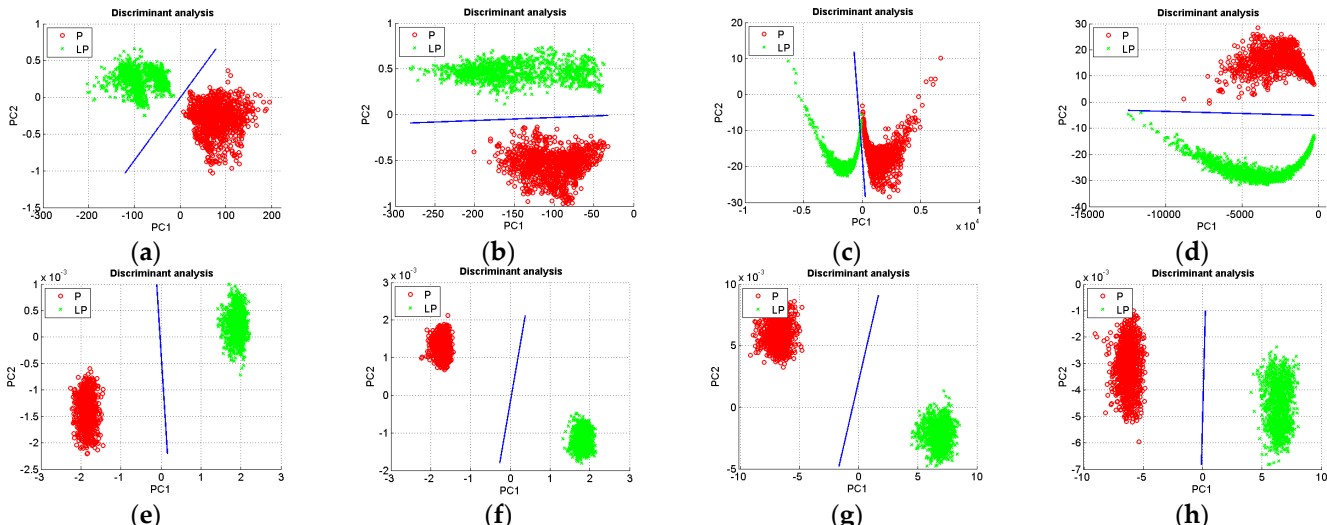

**Figure 6.** Linear discriminant classifier results, general view: (**a**) VIS adaxial, simple; (**b**) VIS abaxial, simple; (**c**) VIS adaxial, polynomial; (**d**) VIS abaxial, polynomial; (**e**) NIR adaxial, simple; (**f**) NIR abaxial, simple; (**g**) NIR adaxial, polynomial; and (**h**) NIR abaxial, polynomial.

Table 6 shows the results of a common classification error using a linear discriminant classifier, in combination with the two methods selected to reduce the amount of data on the spectral characteristics of mulberry leaves. Spectral data from the adaxial and abaxial parts of the leaves, reduced with simple and polynomial variants of kPCA, were compared.

**Table 6.** Common classification error (e, %) for a linear discriminant classifier.

| Method \ Leaf Part Spectra | VIS Adaxial | VIS Abaxial | NIR Adaxial | NIR Abaxial |
|---|---|---|---|---|
| kPC Simple | 0% | 0% | 0% | 0% |
| kPC Polynomial | 1% | 0% | 0% | 0% |

The results show that the lowest values of the common classification error are obtained with the combination of linear discriminant classifier and kernel principal components with "simple" kernel function. For this variant of application, the VIS and NIR spectral characteristics, reduced by the specified method, are defined. The separation functions defined are:

$$\text{VIS adaxial}: \ \delta(PC1, \ PC2) = -0.2099 + [PC1, \ PC2]\cdot[0.155; -18.1305] \quad (16)$$

$$\text{VIS abaxial}: \ \delta(PC1, \ PC2) = 0.0536 + [PC1, \ PC2]\cdot[0.0228; -68.4246] \quad (17)$$

$$\text{NIR adaxial}: \ \delta(PC1, \ PC2) = -5.8636 + [PC1, \ PC2]\cdot\left[-241.6463; -2.022 \times 10^4\right] \quad (18)$$

$$\text{NIR adaxial}: \ \delta(PC1, \ PC2) = 8.3898 + [PC1, \ PC2]\cdot\left[-252.4848; 4.1228 \times 10^4\right] \quad (19)$$

The results obtained indicate that the direct use of the spectral characteristics of the adaxial and abaxial part of leaf petals to passively determine the degree of air pollution in the area of the mulberry habitat is not appropriate. In contrast to the reported results for Tilia leaves by Zadeh et al. [2], spectral indices obtained from the spectral characteristics of the adaxial and abaxial part of the leaves cannot be directly used in mulberry analysis.

The data obtained corroborate those reported by Sun et al. [11], which use spectral characteristics to predict pesticide content in mulberry leaves. For accurate prediction, with an accuracy of 87%, a more sophisticated method of analysis is required, such as regression of the support vectors. In the present work, the separation between leaves

collected from polluted and less polluted areas is obtained after applying the kernel variant of the principal components.

Prediction of water stress in mulberry caused by various factors, including air pollution, was reported by Bhosle et al. [24]. Their proposed method, using REP spectral index, shows a predictive power of 93%. Similar high-resolution accuracy as shown in the present work can be obtained using the averages of these spectral indices. According to the results of other authors and of the results obtained here, it may be recommended to use complex methods of analysis to evaluate changes in mulberry leaves, depending on the pollution of the habitat area of the plant.

## 4. Conclusions

An approach has been adapted to passively determine the degree of pollution by the spectral characteristics of mulberry leaves, based on extracted features and classification. A comparative analysis of the application of methods for reducing the amount of data of spectral characteristics was performed. This analysis found that the direct use of latent variables and the linear variant of principal components is not appropriate in distinguishing between polluted and less polluted areas in an urban environment, according to mulberry leaf data, because the common classification error in using them exceeds 40%.

In the study conducted to determine the degree of air pollution by the spectral characteristics of the mulberry leaf, it was found that this can be realized with a common error of 0–1%, using a linear discriminant classifier, in combination with the kernel variant of the principal components. Analytical dependencies of the separation functions were derived. They were shown to be effective in solving the problem of determining the degree of air pollution in the mulberry habitat.

A strong relationship between the *NDAI* spectral index and chlorophyll was found, due to the fact that the mulberry leaves absorb light most strongly in the blue part of the spectrum, as well as in the red part.

The results obtained improve and complement those reported in the available literature. They can be used to refine the approaches and methods used so far to passively determine the degree of air pollution in the habitat area of the plant.

The proposed methods and software tools could be used in the development of mobile applications and methods for remote measurement, in express determination of the degree of environmental pollution, according to data from the mulberry leaves. More research can be carried out in the subject area, including data on color and spectral indices, as well as combinations of them. Organizing them into feature vectors and processing them with methods to reduce the volume of data would increase the accuracy of forecasting the state of the environment based on data from mulberry leaves.

**Author Contributions:** Conceptualization, S.D. and P.V.-D.; methodology, Z.Z.; software, Z.Z. and P.V.-D.; validation, S.D., P.V.-D. and Z.Z.; formal analysis, S.D.; investigation, Z.Z.; resources, Z.Z.; data curation, P.V.-D. and Z.Z.; writing—original draft preparation, Z.Z.; writing—review and editing, S.D. and P.V.-D.; visualization, Z.Z.; supervision, S.D. All authors have read and agreed to the published version of the manuscript.

**Funding:** This research received no external funding.

**Data Availability Statement:** The Data is available at: https://drive.google.com/file/d/16DuLMoYipRPq2a0V5sbH53cwTZ_u7NK0/view?usp=sharing (accessed on 1 September 2021).

**Conflicts of Interest:** The authors declare no conflict of interest.

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
