# Peer review of "Urban Environmental Quality Assessment by Spectral Characteristics of Mulberry (Morus L.) Leaves"

_environments, doi:10.3390/environments8090087_

Round 1

Reviewer 1 Report

This study explored the feasibility of analyzing the spectral behavior of mulberry leaves. Experiments are conducted which demonstrated the potential. The theoretical background is sound. The analysis method is appropriate. From this reviewer's view, this is a good scientific literature. It may not be novel or with high level innovation. But, this writing should be interesting for some readers.

  1. This reviewer could read this article without problem. But, the writing is a bit awkward. Some paragraphs have only one sentence. It is recommended to have this writing polished by an experienced editor.
  2. It may be the limitation of this reviewer. What is "passive determination"? Is this equivalent to "passive biomonitoring"?
  3. And, what does the description "With existing solutions in this area, the mulberry has been found to be under-researched" mean? Are there any plants other than Tilia (Tilia sp.), hornbeam (Carpinus betulus), white willow, and mulberry? If yes, there would be more plants "under-researched". Why mulberry? And, is mulberry outperforming Tilia and others for passive biomonitoring?
  4. The test area is classified into P and LP. What is the standard of the classification of P and LP?
  5. Could the pollution type be quantified and correlation with the indices be quantitatively analyzed?

Author Response

The answers are included in the attached file. Thank you.

Reviewer 2 Report

   The manuscript by Dineva et al. is devoted to development of new methods of urban environmental quality on basis of spectral measurement. The work is potentially interesting; however, there are questions and comments.

  1. P. 1, lines 34-42: It should be clarified. If plant is tolerance to environmental stressors, then it can weakly respond on these stressors. It means that this plant can be weak bio-indicator.
  2. P. 2, lines 48-52: I am agree that laboratory methods require long time and are invasive. However, these methods are rather more accurate than indirect methods (including spectral methods). Thus, spectral methods are rather high-performance than the most accuracy.
  3. Additionally, the detailed description of widely-used methods of estimation of environmental quality should be included into manuscript. It is also important: What are plants used as bio-indicators of the urban environmental quality?
  4. P. 2, lines 61-64: Plants (and leaves) should be characterized in more detail. In particularly, what was age of investigated plants? What were leaves used? Etc.
  5. P. 2, lines 66-67: Authors used two groups of leaves (from less polluted zones and polluted zones). However, the criteria of the “less polluted zones” and the “polluted zones” should be clarified.
  6. Also, it is not clear: How leaves were transported to laboratory? Was cooling used? What was duration of transporting? Etc.
  7. P. 2, lines 68-69: Spectral measurements should be described in detail. What system was used for spectral measurements? Was it imaging or integral measurement from leaf area? What was light used for leaf illumination? Etc.
  8. Additionally, how much repetitions were used in the work?
  9.   Figure 1: Spectra of leaves from the less polluted zones and the polluted zones seem to be strongly distinguished. Was using complex methods of analysis of spectra necessary?
  10. Authors do not seem to analyze relations of biochemical and physico-chemical characteristics of mulberry to spectral characteristic. The analysis can be basis of quantity estimation of these parameters of basis of spectral characteristics.
  11. Why spectral indices were not analyzed in the work? They can be informative.

Author Response

(The authors gave the same response as above.)

Round 2

Reviewer 1 Report

The authors have responded to the issues raised by this reviewer. No further suggestions.

Reviewer 2 Report

  Authors considered my comments. I have not other questions and remarks.